# Somatic Growth and Maturity for Four Species of River Cooter Including *Pseudemys concinna suwanniensis*, *P. nelsoni*, *P. peninsularis*, and *P. texana*

**DOI:** 10.3390/biology12070965

**Published:** 2023-07-06

**Authors:** Zachary A. Siders, Theresa A. Stratmann, Calandra N. Turner Tomaszewicz, Andrew D. Walde, Eric C. Munscher

**Affiliations:** 1Fisheries and Aquatic Sciences Program, School of Forest, Fisheries and Geomatic Sciences, University of Florida, Gainesville, FL 32605, USA; 2Rewilding Europe, Toernooiveld 1, 6525 ED Nijmegen, The Netherlands; theresastrat@gmail.com; 3NOAA Southwest Fisheries Science Center, 8901 La Jolla Shores Dr, La Jolla, CA 92037, USA; cali.turner@noaa.gov; 4Turtle Survival Alliance—North American Freshwater Turtle Research Group, 1030 Jenkins Road, Suite D, Charleston, SC 29407, USA; awalde@turtlesurvival.org (A.D.W.); emunscher@swca.com (E.C.M.); 5SWCA Environmental Consultants, Department of Natural Resources, 10245 West Little York, Road, Suite 600, Houston, TX 77040, USA

**Keywords:** von Bertalanffy growth, life history invariants, freshwater springs, hierarchical model, sexual dimorphism, environmental effects

## Abstract

**Simple Summary:**

River Cooters commonly occur in many freshwater ecosystems, but little is known about their growth and maturity. We used nearly 20 years of turtle tracking to estimate the growth, length at maturity, age at maturity, and sexual dimorphism of four River Cooters in five freshwater springs, four in Florida and one in Texas. We found that River Cooter juveniles grow exceptionally fast, doubling or tripling in size in the first year, while adults grow very slowly after maturing. Female River Cooters grow larger than males and mature later for the four taxa we studied but much more so for Suwannee River Cooters. Environmental differences among springs affected the growth rate of turtles more than the maximum size. Our efforts elucidate more about the life history of River Cooters and define baseline information to track potential changes to growth from natural and human disturbances in the future.

**Abstract:**

*Pseudemys* is a genus of commonly occurring freshwater turtles with limited growth information across their long lifespans. We used 11,361 mark-recapture events to estimate the somatic growth rates of *P. nelsoni*, *P. peninsularis*, *P. concinna suwanniensis*, and *P. texana* from freshwater springs and developed a Bayesian growth model to estimate the species-specific, site-specific, and individual effects on growth. We corroborated evidence for fast juvenile growth and slower adult growth in *Pseudemys* but found uncommonly fast growth rates, with turtles doubling or tripling in size in the first year. *P. texana* males had the smallest average maximum size ($L_{\infty }$, 243 mm), while *P. c. suwanniensis* females had the largest (423 mm). Environmental conditions at springs had significant effects on k, the growth coefficient, but not $L_{\infty }$. We derived, using a ratio of length at maturity to $L_{\infty }$ (71.7% and 87%, males and females), that females matured 1.15–1.57 times older than males except for *P. c. suwanniensis,* which matured three times older. Given the local abundance declines in many *Pseudemys* from anthropogenic impacts, this study provides important baseline life history information for *Pseudemys* species for use in ongoing conservation efforts and presents a novel hierarchical modeling approach using a long-term mark-recapture dataset.

## 1. Introduction

The genus *Pseudemys* (River Cooter) is a group of freshwater turtles (family Emydidae) distributed throughout the south and eastern United States and northern Mexico [1]. The genus has long been considered a taxonomic “quagmire” and has been the subject of numerous revisions [2]. Currently, there are nine recognized species and subspecies with most species, when found, being locally abundant [3,4,5]. Despite this, little is known about the growth and, subsequently, age at maturity for most of the *Pseudemys* species [3,4,5,6]. With observed and perceived declines in the local abundance of many *Pseudemys* species due to anthropogenic threats [7,8], this critical information on the species’ growth and maturity is imperative for understanding their resilience to anthropogenic threats [9,10]. Here, we focus on providing growth rates for three *Pseudemys* species and one subspecies occurring in Florida and Texas: Florida Red-bellied Cooter (*P. nelsoni*), Suwannee River Cooter (*P. concinna suwanniensis*), Peninsula Cooter (*P. peninsularis*), and the Texas River Cooter (*P. texana*).

The *Pseudemys* genus can be found in a variety of habitats, including large rivers, smaller tributaries, lakes, manmade water features, and freshwater spring habitats [3,4,5,11,12]. In Florida, both the Florida Red-Bellied Cooter and Peninsula Cooter prefer habitats with slow water flow with abundant vegetation [3,5], with Florida Red-Bellied Cooter preferring faster water flows than Peninsula Cooter [12]. The Texas River Cooter and Suwannee River Cooter are more adapted to higher flow systems [4,13,14]. All four species/subspecies occur in freshwater springs and spring run systems [12,14,15,16,17], where higher flows occur in the spring runs, and slower flow areas occur in the spring boils, lagoons, and associated sloughs.

Freshwater springs are excellent for comparing growth across the four *Pseudemys* species/subspecies given that all of these taxa occur in high densities in spring systems [12,14,16,18]. The springs of Florida and Texas are largely limestone-based, which dampens spring water temperature variability resulting in a near-constant temperature reflecting the average annual air temperature. This constant temperature and water flow allows for a year-round growing season for aquatic plants, which provides a constant food source for species of turtles that are strongly herbivorous, like *Pseudemys* [3,4,5]. However, freshwater springs are under increased threat by local to global anthropogenic stressors, including habitat alteration, abundant recreational use, groundwater and recharge zone depletion, pollution, invasive species, and climate change [12,19,20]. Additionally, the water quality and quantity of many springs have decreased drastically over the past several decades, resulting in environmental contrast across spring systems despite their thermal stability [19,20]. Given that these systems are bastions of high local abundance for *Pseudemys*, it is important to establish baseline growth rates in spring systems to monitor potential changes from continued anthropogenic stressors.

A well-documented *Pseudemys* trend has been fast juvenile growth with slower growth (typically < 5 mm/year) following the onset of maturity [3,4,5,6,16]. Only *P. texana* has a formal growth model using length-at-age data, where age was estimated from scute growth annuli [6]. The lack of growth modeling is directly related to the ambiguity in the growth annuli being deposited on an annual schedule [21] and a loss of the growth annuli beyond a few years [4]. As a result, there is high uncertainty in key demographic parameters such as age at maturity and longevity. Generally, females have been noted to grow larger and mature later than males, with Suwannee River Cooter females documented to grow 150% larger [4]. 

With limited length-at-age data, mark-recapture-based models can be a suitable replacement for estimating growth [22,23]. These models are often based on the von Bertalanffy growth model that represents individuals growing towards an asymptotic size [24] and is typically used for taxa exhibiting indeterminate growth. This model assumes all individuals grow identically [22], but this assumption has been relaxed with many modifications [25], most assuming individual variation occurs independently in the asymptotic size, $L_{\infty }$ (see [26] for a review). Another limitation that has been noted is that the von Bertalanffy growth model assumes constant decreasing incremental growth throughout life and cannot accommodate drastic changes in resource allocation [27,28]. In teleosts, this limitation has spurred a series of biphasic growth models generally aimed at addressing changes in the growth rate due to maturation [28]. In testudines, Armstrong and Brooks [23,29] modified the von Bertalanffy to change the Brody growth coefficient, k, at a fixed age at maturation. However, the age at maturation is largely unknown for *Pseudemys* species resulting in a critical assumption and easily resulting in bias if the age at maturity is misspecified or if variation exists between groups (e.g., sex, location, subspecies). 

Here, we make use of long-term, ongoing, mark-recapture efforts at multiple sites of Florida Red-Bellied Cooters, Suwannee River Cooters, Peninsula Cooters, and Texas River Cooters from five freshwater springs systems, four in Florida and one in Texas, to model somatic growth for these taxa. We compare our estimates to the existing sparse literature on *Pseudemys* growth, sexual dimorphism, and the first-year growth increment. We also produce estimates of age at maturity to compare to existing estimates based on limited growth annuli studies. To do such, we make use of life-history invariants that relate the time necessary to reach some proportion of $L_{\infty }$ [30,31,32]. These approaches have been used to derive life history parameters from von Bertalanffy growth extensively in teleosts [33] but recently in data-deficient elasmobranchs [34,35]. For three of the four species, our estimation of growth is the first for the species and represents a major contribution to understanding the life history of these taxa.

## 2. Materials and Methods

### 2.1. Sampling Sites

Wekiwa Springs in Orange County, Florida (Figure 1) is a second magnitude spring that opens up to a large slow-flowing pool (149 × 98 × 6.5 m, length x width x height) then narrows (6–12 m wide) becoming the 1.3 km long Wekiwa Springs Run [36]. Invasive aquatic weeds, principally hydrilla, have been an issue in the past, while algal bloom issues persist [12,16,37]. Volusia Blue Spring in Volusia County, Florida (Figure 1) is a first magnitude spring with a 0.72 km long spring run (30 to 46 m wide) which has been impacted anthropogenically over time through pollution, invasive species, recreational use, and algal growth. The spring acts as the largest manatee (*Trichechus manatus latirostris*) winter refuge on the St. Johns River, and this has left the spring largely denuded of native vegetation [17]. Manatee Springs in Levy County, Florida (Figure 1) is a first-magnitude spring with a public swim area dominated by green algae that becomes a hardwood swamp spring run before eventually emptying into the Suwannee River. The study site also includes several sinkhole ponds, which are connected to the main spring. Fanning Springs (~0.70 ha), located in Levy County, Florida (Figure 1) has historically been considered a first-magnitude spring, but is now a second-magnitude spring. Fanning Springs is the only site in the current study with a presence of motorized boats. The spring area is largely devoid of vegetation, while the spring run is a hardwood swamp. Comal Springs system feeds directly into Landa Lake, located in New Braunfels, Comal County, Texas (Figure 1). American eelgrass dominates large portions of the lake. Depending on water levels, the dense vegetation and algal blooms make parts of the lake become almost impossible to survey [18]. Additional site covariates, including size, discharge, temperature, nitrate levels, vegetation density, algal bloom frequency, and annual visitors, are provided in Table 1.

### 2.2. Sampling Protocol

At each of the five study sites, researchers (a mix of professional biologists, students, and interested citizen scientists) conducted multi-day annual or semi-annual snorkel surveys between 1999 and 2020 led by the Turtle Survival Alliance North American Freshwater Turtle Research Group [38]. For each sampling period, a variable number of snorkelers (typically between 5 and 30, depending on the study site) hand-captured turtles from ca. 0800–1600/1900 h, depending on the time of year and weather conditions. All turtles observed were captured by hand while snorkeling and placed into canoes and brought back to a central location for processing. All turtles were tagged and measured (details below) and then released. For each turtle, we recorded maximum straight-line measurements of carapace length (CL), plastron length, carapace width, and shell height to the nearest 1 mm. We determined the sex of turtles based on secondary sexual characteristics, notably tail length and girth and placement of the cloaca in relationship to the edge of the carapace, according to Ernst and Lovich [39]. We noted any unique features or physical anomalies, such as damage, scars, or coloration for each turtle, which helped confirm individual identity. All turtles were weighed either using hanging Pesola spring scales (Pesola AG, Baar, Switzerland) or Ohaus top-loading digital scales (Ohaus Corporation, Parsippany, NJ, USA), depending on turtle size. All turtles were marked using a variation of the technique described by Cagle [40]. In 2009, passive integrated transponder (PIT) tags were used for turtles with CL greater than 70 mm and injected under the right bridge of the turtle into the inguinal cavity [41]. 

### 2.3. Modeling Growth

The lack of accurate aging structures late into life and the lack of maturation information severely limit the application of simpler modeling approaches to *Pseudemys*. This limitation is due to the nature of mark-recapture growth information, which incorporates individual variation. Without accounting for this variability, growth model parameters have increased uncertainty, and the estimation of population-level growth variation is inflated. Below, we present a modification of the von Bertalanffy growth model to parse the population-level and the individual-level effects on somatic growth. In doing so, we account for the individual variation by using a hierarchical multivariate approach to draw a $L_{\infty }$ and k for each individual. This approach retains strong correlations between $L_{\infty }$ and k [34,42] that previous individual variability approaches focused solely on $L_{\infty }$ have disregarded. Additionally, we pool the mark-recapture datasets across species and sites, which facilitates sharing environmental effects on $L_{\infty }$ and k between similar species monitored across a range of environmental conditions. These density-independent factors are expected to impact somatic growth by changing metabolic rates [43,44]. Pooling of the data facilitates species-specific effects to set the baseline site-specific growth while allowing shared environmental effects on $L_{\infty }$ and k, which is not an unreasonable assumption for this set of species/subspecies given the similarity in habitats they occupy. Without pooling species across sites into one model, it would not be possible to separate out site-specific effects from environmentally driven effects.

#### 2.3.1. Mark-Recapture von Bertalanffy Growth Model

We wished to understand the sex-specific growth of *Pseudemys* spp. across our study sites while accounting for various site covariates that might impact growth. We solely used captures that resulted in zero or positive growth and of individuals that eventually had a sex determined at some point in their capture history. We modified the Fabens [22] implementation of the von Bertalanffy growth model [24] commonly applied to testudines growth modeling [23,29] (Equation (1)).
(1)$${\hat{L}}_{t,i}=L_{\infty }-\left(L_{\infty }-L_{t-1,i}\right)e^{-k\Delta t_{i}}$$
where ${\hat{L}}_{t}$ is the expected length at capture in time t for individual i, $L_{\infty }$ is the asymptotic length (also the average maximum length), $L_{t-1}$ is the length in the past capture event, k is the Brody growth coefficient (the proportion of $L_{\infty }-L_{t}$ obtained per unit time), and Δt is the amount of time between t and t−1 in fractions of a year. We assumed a normal likelihood as has been done in previous testudines growth studies [23,29] (Equation (2)).
(2)$$L_{t,i}\sim N\left({\hat{L}}_{t,i},\sigma \right)$$
where $L_{t,i}$ is the observed length at capture in time t for individual i and σ is the variability in growth around the expected length. 

In order to estimate separate growth parameters across sexes and species/subspecies, we estimated sex- and species-specific growth parameters, $L_{\infty },\;k$, and σ (Equation (3)). We also wished to incorporate individual variability in growth and did so by incorporating a random effect for individual Equations (3)–(5).
(3)$${\hat{L}}_{t,s,j,i}=L_{\infty ,s,j,i}-\left(L_{\infty ,s,j,i}-L_{t-1,i}\right)e^{-k_{s,j,i}\Delta t_{i}}$$
(4)$$\log\;\left(L_{\infty ,s,j,i}\right)=\log\left(L_{\infty ,s,j}\right)+\beta_{i,L_{\infty }}$$
(5)$$\mathrm{logit}\;\left(k_{s,j,i}\right)=\mathrm{logit}\;\left(k_{s,j}\right)+\beta_{i,k}$$
where s denotes species, j denotes sex, i denotes individual, and $\beta_{i}$ is the random effect for $L_{\infty }$ and k. Unfortunately, the nonlinearity of the von Bertalanffy growth model results in growth parameters that are highly correlated [34,42]. We chose to directly estimate this correlation as part of the growth model and structured the individual growth parameter random effects as a set drawn from a multivariate normal distribution (Equation (6)).
(6)$$\left\{\begin{array}{c} \beta_{i,L_{\infty }}\\ \beta_{i,k} \end{array}\right\}\sim MVN\left(\left\{\begin{array}{c} 0\\ 0 \end{array}\right\},\Sigma \right)$$
where the mean of the individual random effects is zero for each transformed growth parameter, and Σ is the covariance matrix of the multivariate normal distribution. For the sake of easing the fitting of this correlation, we decomposed the covariance matrix into the Cholesky factorization (Equations (7) and (8)).
(7)$$\Sigma =diag\left(\tau \right)\Omega diag\left(\tau \right)$$
(8)$$\Omega =W_{\Omega }W_{\Omega }^{\prime }$$
where τ is the scale of the individual random effects, Ω is the correlation matrix between $\log\left(L_{\infty }\right)$ and $\mathrm{logit}\left(k\right)$, and $W_{\Omega }W_{\Omega }^{\prime }$ is the Cholesky factorization of the correlation matrix (it is worth noting that $W_{\Omega }$ is typically written as $L_{\Omega }$, which we replaced to reduce confusion with $L_{t}$) [45].

#### 2.3.2. Incorporating Site Covariates

We also wished to incorporate site covariates into the von Bertalanffy growth model. We did this by modifying Equations (4) and (5) to include the part-worth effects of the site covariates (Equations (9) and (10)).
(9)$$\log\left(L_{\infty ,s,j,i}\right)=\log\left(L_{\infty ,s,j}\right)+\Gamma_{L_{\infty }}X_{g}+\beta_{i,L_{\infty }}$$
(10)$$\mathrm{logit}\left(k_{s,j,i}\right)=\mathrm{logit}\left(k_{s,j}\right)+\Gamma_{k}X_{g}+\beta_{i,k}$$
where Γ is the loading matrix that loads site characteristics, $X_{g}$, at site g on the respective transformed growth parameter. The site covariates we included were site size (hectares), flow (liters per day), temperature (°C), nitrate (mg/L), a categorical variable for vegetation density (low, moderate, high), a binary variable for bloom frequency (low, high), and the number of visitors per year (Table 1).

We simplified this model such that the individual growth parameters incorporated the mean of the transformed growth parameters for each sex, the part-worth site effects, and the individual random effects (Equation (11)).
(11)$$\left[\begin{array}{c} \log\left(L_{\infty ,s,j,i}\right)\\ \mathrm{logit}\left(k_{s,j,i}\right) \end{array}\right]=\left[\begin{array}{c} \log\left(L_{\infty ,s,j}\right)\\ \mathrm{logit}\left(k_{s,j}\right) \end{array}\right]+\Gamma_{\left\{L_{\infty },k\right\}}X_{g}+diag\left(\tau \right)W_{\Omega }diag\left(\tau \right)\beta_{i,\{L_{\infty },k\}}$$

In this formulation, $\beta_{i,\{L_{\infty },k\}}$ vary independently, but the $diag\left(\tau \right)W_{\Omega }diag\left(\tau \right)$ modifies this independence based on the correlation between the transformed $L_{\infty }$ and k. This is equivalent to hierarchical prior on $\beta_{i,\{L_{\infty },k\}}$ that occurs in the multivariate formulation (Equation (6)) but allows vectorization speeding up model fitting. We assumed that the correlation between $L_{\infty }$ and k was shared across species inducing a hierarchical prior on the individual random effects across *Pseudemys* spp. We made this assumption for a practical reason—some species were more heavily sampled across sites than others—and under the broader assumption that closely related species should share similar growth rates [46]. 

#### 2.3.3. Predicted Length at Age and Derived Age at Maturity

Within the model, we desired to estimate the expected length at age and had to assume a size at hatch (birth), $L_{0}$. We collated studies with hatchling sizes to obtain $L_{0}$ for our study’s species, including the mean and standard deviation in $L_{0}$. Several studies were lacking standard errors, and we estimated these standard errors given the mean, range, and assumed quantile of the range (coming from a shared significance level, α). To do such, we built an optimization function that minimized the difference between the probabilities from a normal probability distribution function of the known range and the assumed quantile. We first used this optimization function to estimate a typical significance level, α, for standard errors of hatchling size using a known mean, range, and standard error from the Lindeman [6] study on *P. texana*. This significance level was then used to predict the standard errors for *P. concinna* and *P. nelsoni* from Jackson [47] and Heinrich et al. [48], respectively. The Aresco [49] data on *P. floridana* was used to inform *P. peninsularis* (Table 2). The mean, $\mu_{L_{0}}$, and standard error, $\sigma_{L_{0}}$, of the size at hatch for each species was used to inform the prior (and posterior) distribution on $L_{0}$ (Equation (12)).
(12)$$L_{0,s}\sim N\left(\mu_{L_{0},s},\sigma_{L_{0},s}\right)$$
and this $L_{0}$ was used in the original formulation of the von Bertalanffy growth model [24] to predict the length at age (Equation (13)).
(13)$${\hat{L}}_{t}=L_{\infty }-\left(L_{\infty }-L_{0}\right)e^{-kt}$$
where Equation (13) is analogous to Equation (3) but uses t instead of Δt where t is equal to a vector of ages to predict length at age at, ranging from age zero to 50 and an interval of a tenth of a year.

We also wished to predict length $\left(L_{mat}\right)$, age at maturity $\left(t_{mat}\right)$, and age at capture $\left(t_{cap}\right)$. Life-history invariants [30,32,50] have been used to estimate maturity parameters from the von Bertalanffy growth model by assuming that maturity occurs at some proportion of $L_{\infty }$ [34,35]. Gibbons et al. [51] estimated this proportion for males and females of *Trachemys scripta* for two populations, and we used the average sex-specific proportions of $L_{\infty }$, $\bar{\left(\frac{L_{mat}}{L_{\infty }}\right)}$, for *Pseudemys* (Equation (14)).
(14)$$L_{mat}=\bar{\left(\frac{L_{mat}}{L_{\infty }}\right)}*L_{\infty }$$

We did not find a similar dataset for any *Pseudemys* species, and with the genus *Trachemys* sister to *Pseudemys* [52], it stands to reason that the two genera might mature at similar proportions of $L_{\infty }$. With this length at maturity or the length at capture, we can then derive the expected age at the respective length (Equation (15)).
(15)$$t_{x}=\left(\frac{1}{k}\right)\log\left(\frac{L_{\infty }-L_{0}}{L_{\infty }-L_{x}}\right)\;$$where $t_{x}$ is the age at maturity $\left(t_{mat}\right)$ or the age at capture $\left(t_{cap}\right)$, and $L_{x}$ is the corresponding length. By predicting length at age and estimating a derived $L_{mat}$, $t_{mat}$, and $t_{cap}$ within the model, we ensure that all error in the model estimates of the growth parameters propagates to these predictions/derivations [34,35].

#### 2.3.4. Bayesian Implementation

We fit the mark-capture von Bertalanffy growth model [22] using STAN and the *cmdstanr* package [53]. We specified priors on $L_{\infty }$, k, and σ by estimating Equation (1) for each species and sex using maximum likelihood estimation to estimate the location and setting a coefficient of variation of 50% to determine the respective scale following Caltabellotta et al. [54] and Rolim et al. [34]. We assumed that $\beta_{i,\{L_{\infty },k\}}$ and Γ in Equations (6)–(11) followed standard normal priors. We assumed $W_{\Omega }$ had a prior following the Lewandowski, Kurowicka, and Joe (LKJ) [45] correlation matrix Cholesky factor with a shape of 2 and set a prior on τ following [55]. We fit the model in STAN using four chains with 5000 warmup and 1000 sampling iterations per chain using the NUTS sampler. We assessed chain convergence using the Gelman–Rubin statistic [56].

## 3. Results

### 3.1. Sampling

A total of 421 sampling events yielded 11,361 turtle captures, with 7219 captures of 2469 individual turtles having known sexes and valid length measurements. The median number of captures for individuals ranged from 2–3 across species and sites (range: 1 to 28). We used data from 1987 *P. nelsoni* captures (1049 females and 938 males), 2618 *P. peninsularis* captures (1315 females, 1303 males), 845 *P. c. suwanniensis* captures (409 females, 436 males), 1769 *P. texana* captures (914 females, 855 males). Wekiwa Springs had the highest number of captures, 2368 *P. peninsularis*, and 1736 *P. nelsoni*. Comal Springs had the next highest number of captures, 1769 *P. texana*, Manatee Springs had 542 *P. c. suwanniensis* captured, Blue Springs had 250 *P. peninsularis* and 251 *P. nelsoni* captured, and 303 *P. c. suwanniensis* were captured at Fanning Springs. 

The frequency of sampling varied between sites: 194 events at Wekiwa Springs, 91 events at Comal Springs, 71 events at Blue Springs, 50 events at Manatee Springs, and 30 at Fanning Springs (Figure 1A). Comal Springs had the highest average discovery rate of new individuals (7.2 individuals/trip), with similar rates for Wekiwa Springs, Fanning Springs, and Manatee Springs (3.05–4.09 individuals/trip) and the lowest rates for Blue Springs (0.81–1.95 individuals/trip). We also sampled turtles above the noted maximum size for several species: 330 mm versus 300 mm in Jackson [47] for *P. nelsoni* males, 342 mm versus 320 mm in Thomas and Jansen [5] for *P. peninsularis* males, 349 mm versus 330 mm in Jackson [4] for *P. c. suwanniensis* males, and 344 mm and 294 mm versus 241 mm and 161 mm in Lindeman [6] for *P. texana* females and males, respectively. 

### 3.2. Growth

The mark-recapture von Bertalanffy growth model of species- and site-specific growth rates converged with the Gelman–Rubin statistic below 1.1 for all parameters after 6000 iterations per chain. The median correlation between log-transformed $L_{\infty }$ and logistic-transformed k individual random effects was −0.31 (−0.4−0.23, 90% credible interval). 

#### 3.2.1. Species-Specific Growth Rates

Among the four species modeled, *P. nelsoni* and *P. texana* were the most similar in median female asymptotic size, with $L_{\infty }$ at 311 and 313 mm, respectively, while *P. peninsularis* and *P. c. suwanniensis* females grew to larger sizes, 363 and 423 mm, respectively (Table 3, Figure 2). The asymptotic size of males did not follow this trend as *P. peninsularis* and *P. c. suwanniensis*
$L_{\infty }$ were very similar (309 and 311 mm, respectively) while *P. nelsoni* males had $L_{\infty }$ of 274 mm and *P. texana* males, the smallest of the four species, had $L_{\infty }$ of 243 mm (Table 3, Figure 2). The Brody growth coefficient, k, for *P. c. suwanniensis* males was almost double females and the only species where $k_{M}>k_{F}$ (Table 3, Figure 2). All standard deviations in growth, σ, were similar between males and females (Table 3) except for *P. c. suwanniensis* whose females had the highest variability (Figure 2). The predicted growth increment in the first year, $\Delta L_{a=1}$, for females was highest in *P. peninsularis* (56.9 mm) and similar for *P. texana* (55.2 mm), somewhat lower for *P. c. suwanniensis* (49.5 mm) and lowest in *P. nelsoni* (41 mm). In contrast, the male predicted growth increment in the first year was far greater for *P. c. suwanniensis* (61 mm) and similar among the other three species (31.3–36 mm).

#### 3.2.2. Species-Specific Derived Age at Maturity and Longevity

From Gibbons et al. [51], the ratio of $L_{mat}$ to $L_{\infty }$ ranged from 63.5% to 93%, with a mean ratio of 71.7% for males and 87.2% for females. The result of this ratio is that females mature larger and later than males for all species (Figure 2). Across the study taxa, *P. c. suwanniensis* had the biggest difference between female and male length at maturity ($\Delta L_{mat}$, 145 mm SCL), followed by *P. texana* (99 mm SCL), then *P. peninsularis* (94 mm SCL), and then *P. nelsoni* (74 mm SCL) (Table 3). For *P. peninsularis*, the difference between female and male median age at maturity, $t_{mat}$, was small ($\Delta t_{mat}$, 1.15 years) but was greater for *P. nelsoni* and *P. texana* (3.88 and 2.96 years, respectively) and greatest for *P. c. suwanniensis* (9.79 years). For all species except *P. c. suwanniensis*, this results in females maturing 1.13–1.57 times older than males while, for *P. c. suwanniensis*, females matured 3.25 times older. The oldest median female age at maturity was *P. c. suwanniensis*, 14.12 years, while the youngest was *P. texana*, 8.14 years. The oldest median male age at maturity was *P. peninsularis*, 9.07 years, while the youngest was *P. c. suwanniensis*, 4.36 years (Table 3).

#### 3.2.3. Site-Specific Effects on Growth

None of the site covariates had significant effects, but, when aggregated to the site level, the totality of the environmental conditions had significant effects on k but not $L_{\infty }$ (Figure 3). Among the sites, Fanning Springs had the greatest impacts on $L_{\infty }$ and k, with negative impacts on $L_{\infty }$ and significant positive impacts on k. The remaining sites had negative impacts on k, with all but Manatee Springs having significant effects. Comal Springs and Blue Spring had positive median effects on $L_{\infty }$, while the remaining sites, Wekiwa Springs and Manatee Springs, had slightly negative median effects (Figure 3). The result of these effects on $L_{\infty }$ and k on *P. nelsoni* and *P. peninsularis* was that turtles at Blue Spring had slightly larger asymptotic sizes than Wekiwa Springs and lower k values leading to slightly lower length at maturity in Wekiwa Springs (Figure 4A–D). For *P. c. suwanniensis*, turtles at Manatee Springs had slightly larger asymptotic sizes than Fanning Springs and lower k values leading to slightly lower length at maturity in Fanning Springs (Figure 4E,F). The result is that relative to the species-specific growth rate, the environmentally informed $L_{\infty }$ increased slightly on average, and k decreased on average (Figure 4G).

#### 3.2.4. Sexual Dimorphism

Across the suite of $L_{\infty }$, k, and σ parameters, *P. c. suwanniensis* and *P. peninsularis* had the least overlap between female and male posteriors (2% and 6% on average, respectively), with *P. nelsoni* and *P. texana* having more overlap (24% and 21% on average, respectively) (Figure 2). In the first year of life ($\Delta L_{a=1}$), *P. peninsularis* was the most sexually dimorphic (20% posterior overlap) in SCL, while *P. c. suwanniensis* was the least sexually dimorphic (68%). However, *P. c. suwanniensis* was the most sexually dimorphic (4% posterior overlap) in straight carapace length at their maximum size $\left(L_{\infty }\right)$. These results held between species across sites, but the environmental effects on growth resulted in some within-species site-specific differences in sexual dimorphism. For *P. nelsoni* and *P. peninsularis*, sexual dimorphism in $L_{\infty }$ was greater at Wekiwa Springs (0% posterior overlap for both species) than Blue Spring (47% and 36%, respectively) but similar for k and $\Delta L_{a=1}$. For *P. c. suwanniensis*, Fanning Springs exhibited more sexual dimorphism in $L_{\infty }$ (0% posterior overlap) than Manatee Springs (10%) but no differences in k and $\Delta L_{a=1}$.

## 4. Discussion

Somatic growth is a key life history parameter that correlates with a variety of physiological processes, including maturity, fecundity, and mortality [31,57]. As such, somatic growth is often used as a proxy or surrogate to inform these, at times, more difficult-to-measure processes resulting in somatic growth frequently being critical for understanding a species response to disturbance, natural or anthropogenic [9,10]. Testudines, and more specifically, freshwater turtles, are a group of species whose general life history characteristics, slow-growing, long-lived, late maturing, and low fecundity, put them at risk from human impacts [29,58]. The habitats these species live in are also frequently disturbed, degraded, or destroyed by human activity [59], elevating the need for measuring a species compensatory ability to such impacts. Here, we present somatic growth relationships from a large, multi-site mark-recapture survey of three *Pseudemys* species (*P. nelsoni*, *P. peninsularis*, and *P. texana*) and one subspecies (*P. concinna suwanniensis*). We corroborate the broad finding in *Pseudemys* of rapid juvenile growth and very slow adult growth that begins with the onset of maturity but find exceptionally high juvenile growth rates. The hierarchical Bayesian von Bertalanffy growth model we developed was essential for estimating species-specific growth rates from individual capture histories while accounting for environmental effects on asymptotic length $\left(L_{\infty }\right)$ and growth coefficients $\left(k\right)$ across sites. This method also allowed us to share the uncertainty in the growth parameters with the estimates of length at maturity and age at maturity we derived using life history invariants. Together, the somatic growth rates and maturity information represent a substantial information gain on the life history of these species. 

The somatic growth patterns we estimated broadly corroborate many of the growth findings for past studies on Emydidae species [6,11,16]. For *P. nelsoni*, *P. c. suwanniensis*, and *P. peninsularis*, our somatic growth relationships are the first estimated for the species/subspecies, while for *P. texana* they are the first from mark-recapture surveys. Lindeman [6] estimated a von Bertalanffy growth model for *P. texana* using growth annuli on 26 turtles that were less than six years old, and estimated $k_{M}=0.191,\;k_{F}=0.129$ and $L_{\infty ,M}=$ 169, $L_{\infty ,F}=$ 255 mm SCL, where M and F indicate males and females. We estimated higher k and $L_{\infty }$ values for *P. texana* based on 1769 captures of 914 females and 855 males with $k_{M}=$ 0.201$,\;k_{F}=$0.232 and $L_{\infty ,M}=$ 243, $L_{\infty ,F}=$ 313 mm SCL. There is a chance that the differences in k between the studies are from site-specific differences, as Lindeman [6] sampled the South Llano River, whereas our mark-capture study occurred in the Comal Spring—Landa Lake system. A co-occurring explanation is the Lindeman [6] study lacked older turtles, with their largest female being 274 mm SCL and ours being 344 mm SCL. Poor sampling of larger, older individuals results in lower k as $L_{\infty }$ is overestimated as there is little data to inform the asymptotic size [42]. For *P. concinna concinna*, Dreslik [60] estimated $k_{M}=0.136,\;k_{F}=0.087$ and $L_{\infty ,M}=239,\;L_{\infty ,F}=327$ mm SCL from 15 male and 34 female turtles in Round Pond, Illinois. Similar to the findings for *P. texana*, our estimated *P. c. suwanniensis* growth parameters were considerably higher ($k_{M}=$ 0.258, $k_{F}=$ 0.138 and $L_{\infty ,M}=$ 311, $L_{\infty ,F}=$ 423 mm SCL).

As such, it is perhaps coincident that the first-year growth rates we estimate are exceptional, with females of *P. peninsularis* and *P. texana* as well as males and females of *P. c. suwanniensis* increasing their length more than 49 mm SCL in the first year from hatching size. These rates are high compared to previously published juvenile rates, with Bancroft et al. (1983) estimating 19.3 mm yr^−1^ for *P. nelsoni*, Huestis and Meylan [61] estimating 35 mm yr^−1^ for *P. c. suwanniensis*, and Gibbons and Coker [62] estimating 13 mm yr^−1^ for *P. peninsularis*. For female *P. texana* and male *P. c. suwanniensis*, our growth rate estimates result in a one-year-old turtle growing to over 100 mm SCL and already close to a third of its asymptotic size. *P. c. suwanniensis* standouts with the highest growth coefficient among the study species with one-year-old males predicted to grow almost 12 mm SCL larger than females in the first year, the only species in our study to do so. Males in the other *Pseudemys* species/subspecies were typically 58–76% of female size in the first year of life. For *P. nelsoni* and *P. texana*, males and females had similar growth curves with smaller differences between the sex-specific k and $L_{\infty }$, while *P. peninsularis* males exhibited the slowest growth among the males and females of all the species/subspecies in our study. These elevated growth rates are potentially due to the spring systems we sampled. Springs provide a thermally stable environment reducing the effects of seasonality on primary production and providing turtles with a consistent food supply year-round. Huestis and Meylan [61] found elevated average growth rates (35 mm yr^−1^) for *P. c. suwanniensis* in Rainbow Run, another spring system in Florida. 

We found strong evidence for sexual dimorphism among the species/subspecies, with females larger than males, similar to past studies. Sexual dimorphism was the most extreme in *P. c. suwanniensis,* with the $L_{\infty }$ of females 1.36 times greater than that of males, which is similar but lower than the Jackson [4] estimate of 1.5 times. Interestingly, the smallest species among the set, *P. texana*, had the next highest sexual dimorphism with a ratio of $L_{\infty ,F}/L_{\infty ,M}$ of 1.29. Both *P. nelsoni* and *P. peninsularis* had similar ratios (1.14 and 1.17, respectively) but were consistent with past studies which found females were larger than males [5,63]. This results in median sexual dimorphism index (SDI; [64]) of 0.135, 0.174, 0.36, and 0.288 for *P. nelsoni*, *P. peninsularis*, *P. c. suwanniensis*, and *P. texana*, respectively, based on $L_{\infty }$. These differ from previous studies: for *P. peninsularis*, this is far lower than Aresco’s [49] SDI for *P. floridana* of 0.50, for *P. c. suwanniensis*, our SDI is slightly lower than that of Jackson and Walker’s [65] SDI of 0.45, for *P. texana*, this is very different from the Lindeman [6] estimate of 0.88, though Lindeman notes their SDI is likely skewed by sampling. In all, our study quantified long-observed similarities between the *Pseudemys* species and subspecies we sampled while corroborating some of the observed differences and providing evidence for some additional differences that have not been widely noted. 

By using life-history invariants of somatic growth, we were able to derive estimates of length- and age-at-maturity for the *Pseudemys* species/subspecies in our study. Given that these quantities are not estimated from direct measurements of maturity, we chose to generate these quantities within the model to pass any uncertainty in the growth parameters onto the uncertainty of the length- and age-at-maturity estimates. To facilitate this derivation, it was necessary for us to generate a percent of $L_{\infty }$ that length at maturity occurs, $L_{mat}$:$L_{\infty }$. We calculated this ratio as 71.7% and 87.2% of $L_{\infty }$ for males and females, respectively, from *Trachemys scripta* from measurements of maturity and estimates of $L_{\infty }$ made by Gibbons et al. [51]. These are higher than the ratio commonly used in teleosts, 66%, but smaller than has been used for sea turtles, 97.5% [32,66]. It should be noted that this ratio is a fundamental assumption of our derived length at maturity and age at maturity estimates, and we shared this ratio across species. Future studies should focus on generating this life-history invariant ratio of length at maturity to asymptotic length for a variety of *Pseudemys* and other turtle species.

For *P. nelsoni*, Bancroft et al. [7] estimated males mature at three years old and females mature at seven to eight years old. We estimated a far higher male age at maturity (8.26 years) but only a slightly higher estimate for females (12.2 years). However, our lengths at maturity, 197 mm SCL for males and 271 mm SCL for females, are consistent with the reported length at maturity for *P. nelsoni* between 187–231 mm SCL for males and 275–290 mm CL for females [47]. In *P. c. suwanniensis*, Bancroft et al. [7] estimated males mature at five years and females at 16 years, while Jackson and Walker [65] estimated females mature at 10–13 years. Our estimates were close to Bancroft et al. [7] for males at 4.36 years and between the two studies for females at 14.12 years. Additionally, our lengths at maturity, 223 mm SCL for males and 369 mm for females are consistent with past report values of 216 mm and 360 mm SCL [4]. For *P. peninsularis*, Congdon and Gibbons [67] estimated males mature at three to four years and females mature at five to seven years, while we estimated far higher age at maturity, with males at 9.07 years and females at 10.25 years. Coincidentally, we estimate higher lengths at maturity, 222 mm SCL for males and 316 mm SCL for females, than previously reported 130–150 mm SCL for males and 240–300 mm SCL for females [62,67]. While Lindeman [6] does not report age at maturity for *P. texana*, the reported length of mature individuals were 87–178 mm SCL for males and 234–265 mm SCL for females, with our reported lengths at the high end for males (174 mm SCL) and above the maximum for females (273 mm SCL).

The unexpected late age at maturity of males for *P. nelsoni* and *P. peninsularis* likely results from separate phenomena. In *P. nelsoni*, the agreement between our estimated lengths at maturity and those published from observed maturation indicates that the ages estimated in Bancroft et al. [7] are underestimated. In *P. peninsularis*, it is likely that this comes down to the split of *P. floridana* where Congdon and Gibbons [67] sampled *P. floridana* in South Carolina, whereas we sampled *P. peninsularis* (syn. *P. floridana peninsularis*) in peninsular Florida [1]. Another potential source of discrepancy is that the $L_{mat}$ to $L_{\infty }$ ratio was derived from the Gibbons et al. [51] study of *T. scripta* and, without a study of the variability in life-history invariants across Deirochelyinae or Emydidae, there is some reason to interpret our $L_{mat}$ and $t_{mat}$ with circumspection. However, the close agreement between many of the lengths at maturity across our study species suggests that the Gibbons’ et al. [51] invariants are likely appropriate for *Pseudemys*. Life-history invariants have been useful for understanding the variation of life-history strategies across taxa [31,32], and future studies should be pursued in Testudines to understand how these invariants may differ from other indeterminate growing clades.

The joint von Bertalanffy growth model made some key assumptions. Principally, we assumed the environmental effects on $L_{\infty }$ and k were shared among species. Ideally, we could relax this assumption with further overlap between species and sites (or with additional sites) but with at most any species occurring at two sites, we would have suffered from separability issues in the current model. We concluded this pooling of effects was appropriate as no one species at a site massively outnumbered another in terms of captures, and we still resolved significant site-specific effects on k. Interestingly, we did not estimate any significant effects by any given environmental covariate on $L_{\infty }$ or k but solely when in aggregate at a site (Figure 3). We attribute this to the variability in individual growth as well as the low overlap between species and sites.

We also made a structural assumption that individual variation occurs in $L_{\infty }$ and k, which differs from past approaches that assume individual variation only occurs in $L_{\infty }$ (see [26] for a review). We made this assumption for biological, empirical, and structural reasons. Growth is the net result of anabolic (energy gain) and catabolic (energy loss) processes, and the von Bertalanffy model [24,68] formulates k parameter to depend on the catabolic rate while the $L_{\infty }$ depends upon the ratio of anabolism and catabolism. However, there is some debate on how von Bertalanffy [24,68] defines catabolism and whether catabolism includes processes related to routine behaviors such as obtaining food [43]. Empirically, the spring systems studied here are remarkably thermally stable [36], reducing any direct temperature-related effects on k as have been shown in other studies [44]. We were additionally unwilling to discern whether a specific environmental covariate would solely affect anabolic or catabolic processes as none directly measured density-dependent effects or food availability. Structurally, $L_{\infty }$ and k are highly correlated parameters demonstrated by our negative correlation of −0.31 even in transformed space. Allowing for individual or environmental variation in one parameter while fixing the other seemed unlikely to garner robust growth curves. Failing to estimate one parameter properly has been shown to have severe, negative biases on the corresponding parameter [42]. Thus, we chose to let the data decide where environmental covariates played a role, in $L_{\infty }$ or k.

## 5. Conclusions

Given these assumptions, the novel estimates of *Pseudemys* species generated with the hierarchical Bayesian somatic growth model provide critical information for the ongoing management and conservation of these similar species [2,52]. Somatic growth rates and age at maturity in wild populations are major data gaps for many freshwater turtle species preventing a robust understanding of the ability to compensate for anthropogenic impacts [3,4,5,58]. Thus, increasing the understanding of how freshwater turtles grow can aid in management of the species at local, state, regional, and range-wide levels [9]. Changes in growth rates at a population level can act as an early warning sign for changes in environmental conditions and or impacts on habitat, such as water quality, food availability, and climate change. Our study serves as a baseline reference for the species, and at these impacted spring sites and provides a benchmark to track the population into the future. These benchmarks are essential for measuring restoration or management intervention successes in the highly impacted spring systems these species inhabit [20].

## Figures and Tables

**Figure 1 biology-12-00965-f001:**
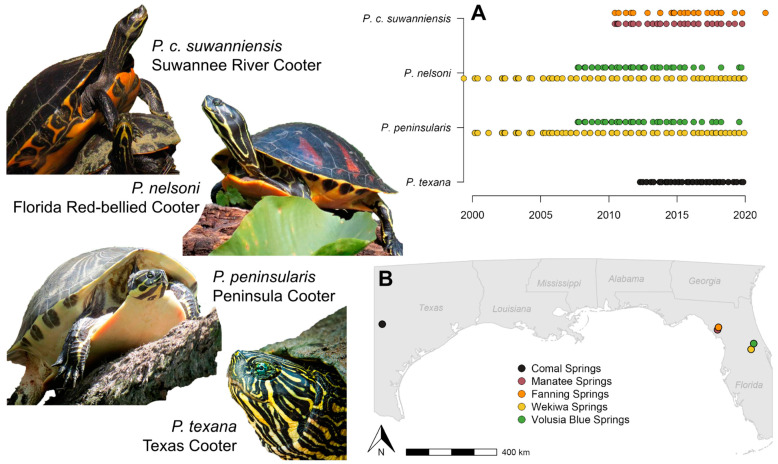
(**A**) Mark-recapture sampling dates for the four River Cooters (*Pseudemys concinna suwanniensis*, *P. nelsoni*, *P. peninsularis*, *P. texana*) from the five spring systems (**B**) Map showing the location of the five study springs: Comal Springs, TX; Manatee Springs, FL; Fanning Springs, FL; Wekiwa Springs, FL; and Volusia Blue Springs, FL.

**Figure 2 biology-12-00965-f002:**
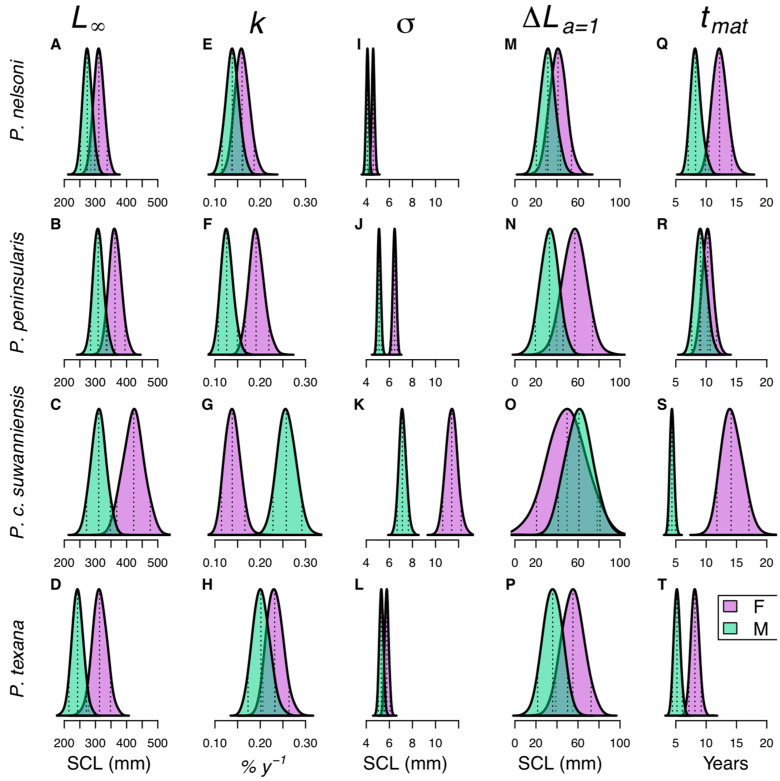
Posterior distributions of species-specific asymptotic length ($L_{\infty }$; (**A**–**D**)), Brody growth coefficient (k; (**E**–**H**)), growth variability (σ; (**I**–**L**)), the growth increment in the first year ($\Delta L_{t=1}$; (**M**–**P**)), and age at maturity ($t_{mat}$; (**Q**–**T**)) parameters for *Pseudemys nelsoni* (**A**,**E**,**M**,**Q**), *P. peninsularis* (**B**,**F**,**J**,**R**), *P. concinna suwanniensis* (**C**,**G**,**K**,**S**), and *P. texana* (**D**,**H**,**L**,**T**). Shaded regions indicate the full posterior distribution, while dashed lines indicate the 5%, 50% (median), and 95% quantiles for females (purple) and males (green) of each species. Note that age at maturity is derived from life history invariants (see Section 2.3.3).

**Figure 3 biology-12-00965-f003:**
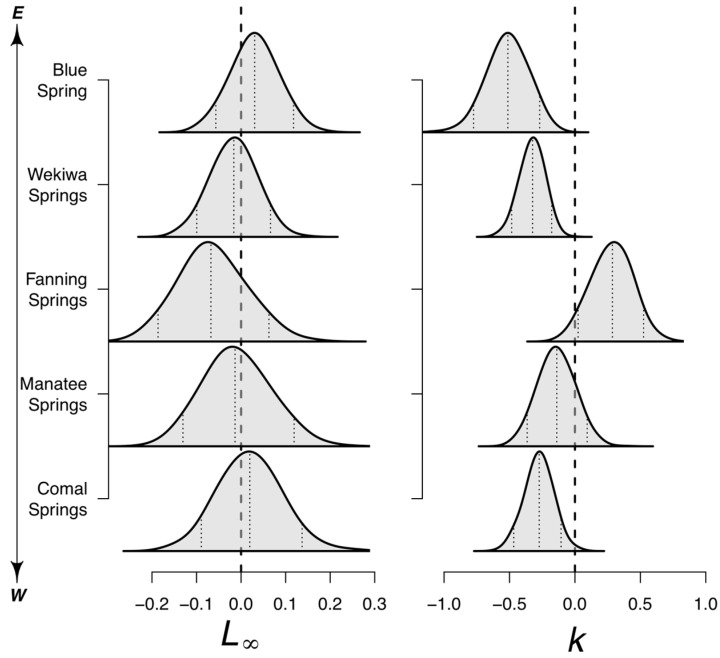
Site-specific effects on the asymptotic length ($L_{\infty }$; **left**) and the Brody growth coefficient for four *Pseudemys* species (k; **right**). These effects are shared among species sampled at a site and added to the log-transformed $L_{\infty }$ or logit-transformed k, respectively, to generate site and species $L_{\infty }$ or k.

**Figure 4 biology-12-00965-f004:**
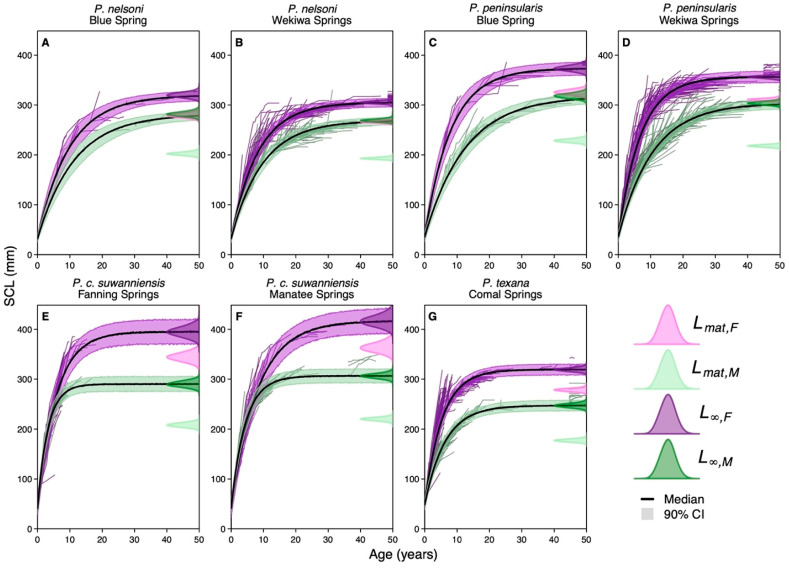
Predicted length at age for female (purple) and male (green) *Pseudemys* species sampled at five spring systems. For each species and site, the median (solid line) and 90% credible interval of the prediction (shaded region) are shown from zero to fifty years and are overlaid on top of the back-calculated time series of length at age for each individual’s capture history. On the right side of each plot is the posterior distribution of the species and site-specific asymptotic length ($L_{\infty }$, dark shaded region) and length at maturity ($L_{mat}$, light shaded region). Note that length at maturity is derived from life history invariants (see Section 2.3.3).

**Table 1 biology-12-00965-t001:** Site covariates of size (hectares), flow (millions L day^−1^), temperature (Temp., °C), nitrate levels (mg/L), categorical vegetation density, categorical algal bloom frequency (Bloom), and visitors per year (Visits) for the five sampling locations.

Site	Size	Flow	Temp.	Nitrate	Vegetation	Bloom	Visits
Wekiwa Springs State Park	2.67	164	22	1.2	Moderate	High	325,000
Blue Springs State Park	1.9	394	23	0.7	Low	Low	485,000
Manatee Springs State Park	1.53	380	22	2	Moderate	High	174,000
Fanning Springs State Park	0.7	300	22	6	Low	High	180,000
Comal Springs–Landa Lake	8.4	726.8	21	1.8	High	High	1,000,000

**Table 2 biology-12-00965-t002:** The mean $\left(\mu \right)$, standard deviation $\left(\sigma \right)$, and range of the size at birth $\left(L_{0}\right)$ for each *Pseudemys* species and the source. Standard deviations noted with asterisks were estimated given the species’ mean and range, and assuming an α equal to that would result in the range for *P. texana* given its mean and standard deviation.

Species	*μ*	*σ*	Range	Source
*P. nelsoni*	32.1	1.84 *	29–35	Jackson [47]
*P. peninsularis*	35.6	2.50	—	Aresco [49]
*P. c. suwanniensis*	41.3	1.02 *	40–43	Heinrich et al. [48]
*P. texana*	48.0	3.61	41–53	Lindeman [6]

**Table 3 biology-12-00965-t003:** von Bertalanffy growth parameters $\left(L_{\infty },\;k,\;\sigma ,\;t_{0}\right)$ for the four *Pseudemys* taxa in this study as well as the first-year growth increment $\left(\Delta L_{a=1}\right)$ and the age at maturity $\left(t_{mat}\right)$. The median and 90% credible interval (in parentheses) are reported for each parameter. Note that $t_{mat\;}$ was derived from the life-history invariants and was not directly informed by maturity measurements.

Species	θ	Females	Males
*P. nelsoni*	$L_{\infty }$	311 (286–338)	274 (252–298)
	k	0.16 (0.137–0.186)	0.138 (0.116–0.161)
	σ	4.6 (4.41–4.81)	4.1 (3.92–4.28)
	$\Delta L_{a=1}$	41 (29.5–53.9)	31.3 (20.3–43.3)
	$L_{mat}$	271 (249–294)	197 (181–214)
	$t_{mat\;}$	12.2 (10.46–14.21)	8.26 (7.04–9.89)
	$t_{0\;}$	−0.681 (−0.842–−0.55)	−0.902 (−1.12–−0.733)
*P. peninsularis*	$L_{\infty }$	363 (334–395)	309 (284–337)
	k	0.191 (0.167–0.219)	0.126 (0.108–0.149)
	σ	6.45 (6.21–6.68)	5.12 (4.91–5.34)
	$\Delta L_{a=1}$	56.9 (39.8–73.7)	32.8 (19.4–45.7)
	$L_{mat}$	316 (291–344)	222 (204–242)
	$t_{mat\;}$	10.25 (8.86–11.69)	9.07 (7.63–10.64)
	$t_{0\;}$	−0.538 (−0.657–−0.444)	−0.969 (−1.21–−0.777)
*P. c. suwanniensis*	$L_{\infty }$	423 (368–476)	311 (272–348)
	k	0.138 (0.115–0.166)	0.258 (0.228–0.292)
	σ	11.4 (10.6–12.2)	7.13 (6.68–7.65)
	$\Delta L_{a=1}$	49.5 (20.5–78.3)	61 (41.2–81)
	$L_{mat}$	369 (321–415)	223 (195–250)
	$t_{mat\;}$	14.12 (11.78–17.04)	4.36 (3.85–4.92)
	$t_{0\;}$	−0.746 (−0.949–−0.594)	−0.555 (−0.675–−0.463)
*P. texana*	$L_{\infty }$	313 (278–348)	243 (215–271)
	k	0.232 (0.206–0.264)	0.201 (0.173–0.233)
	σ	5.78 (5.53–6.06)	5.3 (5.04–5.58)
	$\Delta L_{a=1}$	55.2 (38.9–72.5)	36 (21.6–50.2)
	$L_{mat}$	272 (242–304)	174 (154–195)
	$t_{mat\;}$	8.14 (7.16–9.18)	5.19 (4.44–6.06)
	$t_{0\;}$	−0.722 (−0.889–−0.58)	−1.1 (−1.38–−0.866)

## Data Availability

The data and code that support the findings of this study are available from the corresponding author, [ZS], upon reasonable request.

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
