# Peer review of "Somatic Growth and Maturity for Four Species of River Cooter Including Pseudemys concinna suwanniensis, P. nelsoni, P. peninsularis, and P. texana"

_biology, 2023, doi:10.3390/biology12070965_

Round 1

Reviewer 1 Report

We didn't find any major comments. We have only proposed a reorientation of the last paragraph of the introduction, which we feel should be more of a methodology than an introduction. There are also a few errors to harmonize the bibliography of the manuscript.

Author Response

We thank the reviewer for the comment. We have reorganized the last paragraph of the introduction to better contextualize the methodological components of the paragraph and make the paragraph more introductory style. 

Reviewer 2 Report

Very interesting and relevant research! The authors have conducted a lot of field research, the methods used are relevant and effective. The processing of the results is correct and the conclusions are substantiated. The results of the study will be of interest both to scientists studying freshwater turtles and to the practical protection of freshwater turtles and their habitats.

Author Response

We thank the reviewer for their thoughtful comments.

Reviewer 3 Report

General comments:

This is an interesting manuscript which presents new and valuable data on freshwater turtle growth and age at maturity. More specifically, the authors use an impressive long-term dataset across four species to disentangle species-specific differences from site-specific determinants of the measured parameters. The data appear overall well analysed, and the conclusions appear supported by the presented evidence. The display items are particularly nicely prepared.

While the MS clearly deserves publishing, the following two general points caught my eye:

(i) As a whole, the manuscript is written in a rather ‘inward-looking’ rather than ‘outward-looking’ way. It presents a very large amount of details both from a methodological-technical as well as from a species-biological view, while the wider context of the work goes a little missing. As a main rationale and aim, the authors refer to the provision of baseline information for further demographic population monitoring in a conservation context. However, after going through the detailed models and their detailed interpretation, many readers might be left wondering whether more straightforward, simpler modelling approaches would also result in the required information for such a monitoring scheme.

I am aware that this is not an overly constructive comment, as it does not offer a straightforward solution to improve the manuscript. Maybe the authors could consider (i) shortening the manuscript, (ii) move some rather technical parts to supplements, or (iii) tweak the manuscript throughout to move the most important take-home messages more to the foreground.

(ii) Formally, the manuscript appears a little insufficiently prepared for the target journal – giving the impression that some of the formatting might stem from a previous submission elsewhere. Some of this is listed in the specific comments below.     

Specific comments:

- General: In most scientific journals, small letters are used for common species names unless the common name consists of e.g. a locality (‘river cooters’ rather than ‘River Cooters’). Maybe the author instructions or journal editors could give further advice on the matter.

- The Abstract might need re-formatting, either by removing the numbering system (more common for Biology?) or by starting a new paragraph with each number

- Line 34: explain the mathematical symbols to make the Abstract self-explanatory.

- Line 134-: I got a little confused by 2.1. Sampling Protocol being shown before 2.2 Sampling Sites. In my mind, a site description is a pre-requisite to understand the sampling protocol, and the authors should consider swapping 2.1. with 2.2.

- Line 430 (and also elsewhere): use the numbering system for references throughout

- Lines 592-593: remove the general instructions (provided by the journal?)

- The reference list needs another general formal overhaul, and contains myriads of little inconsistencies (italic letters for scientific names, use of capital letters for journal names, etc.), Also journal article titles should generally not use capitalized words (an example from Reference 2: ‘Misleading Phylogenetic Inferences Based on…..’ should be ‘Misleading phylogenetic inferences based on….’).

Author Response

We thank the reviewer for their thoughtful comments. We agree with the main takeaway that we did not make it abundantly clear why undertake such a technical approach. We have opted for option (iii) the reviewer suggested by tweaking the manuscript throughout to make the main take home messages more clear. This was principally done in the last paragraph of the introduction and the discussion.

General: The lead author comes from the American Fisheries Society world where common names are capitalized, so this is a legacy of that. 

Abstract: Removed numbering.

Line 34: Added mathematical symbols to make equations self-explanatory

Line 134: Agree, reordered.

Line 430: Fixed.

Line 592-593: Fixed.

References: Fixed.